# L-Serine Influences Epigenetic Modifications to Improve Cognition and Behaviors in Growth Hormone-Releasing Hormone Knockout Mice

**DOI:** 10.3390/biomedicines11010104

**Published:** 2022-12-30

**Authors:** Fang Zhang, Mert Icyuz, Trygve Tollefsbol, Paul Alan Cox, Sandra Anne Banack, Liou Y. Sun

**Affiliations:** 1Department of Biology, University of Alabama at Birmingham, Birmingham, AL 35254, USA; 2Structural Biology Program, Memorial Sloan Kettering Cancer Center, New York, NY 10065, USA; 3Brain Chemistry Labs, Institute for Ethnomedicine, Jackson, WY 83001, USA

**Keywords:** growth hormone-releasing hormone, anxiolytic drug candidates, cognitive improvement, epigenetic modifications

## Abstract

Neurodegenerative diseases feature changes in cognition, and anxiety-like and autism-like behaviors, which are associated with epigenetic alterations such as DNA methylation and histone modifications. The amino acid L-serine has been shown to have beneficial effects on neurological symptoms. Here, we found that growth hormone-releasing hormone knockout (GHRH-KO) mice, a GH-deficiency mouse model characterized by extended lifespan and enhanced insulin sensitivity, showed a lower anxiety symptom and impairment of short-term object recognition memory and autism-like behaviors. Interestingly, L-serine administration exerted anxiolytic effects in mice and ameliorated the behavioral deficits in GHRH-KO. L-serine treatment upregulated histone epigenetic markers of H3K4me, H3K9ac, H3K14ac and H3K18ac in the hippocampus and H3K4me in the cerebral cortex in both GHRH-KO mice and wild type controls. L-serine-modulated epigenetic marker changes, in turn, were found to regulate mRNA expression of *BDNF*, *grm3*, *foxp1*, *shank3*, *auts2* and *marcksl1*, which are involved in anxiety-, cognitive- and autism-like behaviors. Our study provides a novel insight into the beneficial effects of L-serine intervention on neuropsychological impairments.

## 1. Introduction

Neuropsychiatric disorders, such as anxiety, autism and cognitive deficits, are health challenges in humans throughout their lifespan [1]. DNA methyltransferase (Dnmt) [2], Ten-eleven translocation (Tet) [3] and histone H3 modifications in lysine residues [4] are associated with behavioral changes in mammals during aging. In addition, DNA methylation and histone modifications regulate gene expression, such as brain-derived neurotrophic factor (*BDNF*) [5], forkhead box p (Foxp1) [6], SH3 and multiple ankyrin repeat domains 3 (*Shank3*) [7], autism susceptibility candidate 2 (Auts2) [8], myristoylated alanine-rich C-kinase substrate like 1 (Marcksl1) [9], and function in anxiety-like, autism-like and cognitive phenotypes.

Growth hormone (GH) plays a pivotal role in somatic growth, lifespan, metabolism and other endocrine effects. Mice with mutations in GH signaling resulting in GH deficiency or resistance are characterized by extended lifespan, enhanced insulin sensitivity, and delay of` age-related diseases [10]. The aged GH deficient Ames dwarf *(Prop*^df/df^) mice have been shown to maintain better cognitive function and locomotion than control mice [11]. Growth hormone receptor knockout (GHR-KO) mice have higher total number of neurons [12] and improved learning and memory compared with age-matched wild type controls [13]. Age-related decline in learning and memory is delayed and anxiolytic behavior is found in growth hormone-releasing hormone knockout (GHRH-KO) male mice [14,15]. In young rhesus monkey studies, low GH responsiveness was correlated with anxious behaviors [16]. Long-term treatment with GH improves mood and memory in patients with childhood onset GH deficiency [17]. Furthermore, the administration of GH or GHRH attenuates age-related decline in spatial and reference memory in rats [18,19].

L-serine plays a critical role in central nervous system development, neuronal signaling and synaptic plasticity [20]. Oral dosing of amyotrophic lateral sclerosis (ALS) patients with L-serine at 30 g per day slowed disease progression and similar dose regime showed efficacy against a hereditary type of neuropathy [21]. Diet supplemented with L-serine improves cognitive performance and communication in patients with severe neurodevelopmental disorder-induced Rett syndrome-like phenotype [22], reduces the production of neurotoxic deoxysphingolipids resulting in improved motor and sensory performance in hereditary sensory autonomic neuropathy type 1 [23], and lowers homocysteine concentrations, thus reducing the effects of alcoholic fatty liver disease in mice and rats [24]. L-serine intervention decreased β-N-methylamino-L-alanine-triggered neurofibrillary tangles, which is the neurological hallmark of Alzheimer’s disease, in nonhuman primates [25] and reduced the number of protein inclusions in anterior horn neurons along with the amount of reactive gliosis in a vervet model of ALS/MND [26].

In this study, we found that disruption of GH signaling decreased anxiety symptoms, however, caused impairment of short-term object recognition memory and autism-like behaviors in adolescent females. Interestingly, L-serine administration exerted anxiolytic effects in mice and ameliorated the behavioral deficits in KO mice. Furthermore, L-serine treatment upregulated the gene expression of H3K4me, H3K9ac, H3K14ac and H3K18ac in the hippocampus and H3K4me in the cerebral cortex in both KO and WT mice. In turn, the mRNA level of *BDNF*, *grm3*, *foxp1*, *shank3*, *auts2* and *marcksl1* was regulated, which are involved in anxiety-, cognitive- and autism-like behaviors. Our study provides a novel insight into the beneficial effects of L-serine treatment in the amelioration of short-term memory impairment, anxiety- and autism-related behavioral deterioration.

## 2. Materials and Methods

Mice were housed in the Association for Assessment and Accreditation of Laboratory Animal Care–accredited Animal Resources Program facility at the University of Alabama at Birmingham, in accordance with procedures of the Animal Welfare Act and the 1989 amendments to the Act, and all studies followed protocols approved by the University of Alabama at Birmingham Institutional Animal Care and Use Committee.

### 2.1. Antibody and Chemicals

Antibodies for H3K4me and H3K27me were from Abcam (Cambridge, MA, USA); Abs for H3K9ac, H3K14ac, H3K18ac, H3K27ac, Histone H3 and GAPDH were from Cell Signaling Technology (Danvers, MA, USA); protease inhibitor tablet, phosphatase inhibitor tablet, HRP-linked anti-rabbit and HRP-linked anti-mouse secondary Antibodies, Western blot stripping buffer and PowerUp SYBR green master mix for real-time quantitative PCR were from Thermo Fisher Scientific (Waltham, MA, USA). RNeasy plus kit was from Qiagen (Hilden, Germany). LunaScript RT SuperMix Kit was from NEB (Ipswich, MA, USA), L-serine supplement was purchased from BULKSUPPLEMENTS.COM (Henderson, NV, USA) (Table 1).

### 2.2. Mouse Model and L-Serine Administration

As previous described [27], the GHRH-KO mice were generated using CRISPR/Cas9 technology and maintained on a mixed genetic background to avoid any possible strain-dependent phenotypes. GHRH-KO and WT controls were maintained with ad libitum access to food and drink with 12 h light-dark cycles. Six-month-old female mice were fed by water with the addition of 10% L-serine (pH adjusted) as the treatment group, while water without L-serine supplementation was used as the vehicle control. Animals were given 10% L-serine for 12 weeks until they were used for behavioral tests after which the mice were euthanized for tissue harvesting. Water intake was measured weekly.

### 2.3. Animal Behavioral Tests

The mice with the home cage were transported to the test room before the behavioral testing began 30 min. For the entire test, the room temperature and equipment were maintained at 22 °C. In addition, the lighting was kept dim and the light source was not directed towards the experiment area.

### 2.4. Elevated Zero Maze (EZM) and Open Field Test (OFT)

EZM was an elevated cycling corridor with two open areas and two closed areas [28]. Open field trials were conducted in an area of 42 cm^2^ [29]. The data were acquired using the automated EthoVision camera-driven tracker system (Noldus, The Netherlands). To avoid variation in mobility resulting from body size between WT and KO mice, total time spent and entry in open/close arms of EZM and center/side of the OFT were analyzed. The ratio of entry in open arms of EZM was analyzed by entries in open arms versus entries of open and enclosed arms. The ratio of entry in center of the OFT was determined by entries in center versus entries of center and sides.

### 2.5. Novel Objective Recognition (NOR)

NOR was performed as described [30] with modifications. The mice were habituated to the test in an empty open field arena of 42 cm^2^ for 1 h, the day before each NOR test. 24 h after habituation, each mouse was allowed to explore the familiar arena with two identical objects placed at an equal distance for 10 min. After the training session, the mice were returned to their home cage for 4 h. During the testing session, the mice were reintroduced to the open field arena with an original (familiar) object and one novel object to record exploration time and entries of each object individually using the automated EthoVision camera-driven tracker system (Noldus, The Netherlands). Object exploration was defined as a touching between the mouse’s nose and the object or approaching the object at less than 2 cm distance with the mouse oriented toward the object. The percentage of exploration time was calculated using percentage of time spent exploring novel or familiar objects divided by the total exploration time. The frequency of cognition in novel and familiar objects was recorded, respectively.

### 2.6. Three-Chamber Sociability

The size of three-chambered cage was 40 × 20 × 26 cm (W × H × D) with a center chamber of 12 cm (W) and side chambers of 14 cm (W). Both side chambers contained a metal meshed cage in the corner with a weight on it. The assay consisted of two sessions. The first session allowed the mouse to freely explore all three chambers for 10 min in the empty box. After habituation, the mouse was then gently confined in the center chamber while a stranger mouse was placed in the metal meshed cage of the side chamber. The subject mouse was then allowed to freely explore all three chambers for 10 min. During the sociability session, the location of the stranger mouse was switched in a pseudorandom order between side chambers within groups to prevent side preference. The stranger mouse was 6-m-old female wild type and had never-before-housed with the experimental mice. All data were recorded with the automated EthoVision camera-driven tracker system (Noldus, The Netherlands). The percentage of exploration time was calculated using percentage of time spent exploring empty chambers or the stranger mouse chamber divided by the total exploration time. The frequency of exploration in the empty cage and mouse cage were recorded, respectively.

### 2.7. RNA Extraction and Real-Time PCR

RNA was extracted from the cerebral cortex and the hippocampus using RNeasy plus kit. RNA was reverse-transcribed with LunaScript RT SuperMix. Real-time quantitative PCR was performed in Applied Biosystems QuantStudio 3. PCR reactions contained 25 ng of cDNA template and 10 nM of primers in a final volume of 10 μL of SYBR green master mix. Expression of β-actin and GAPDH was used to normalize gene of interest in each sample [27]. The following primers were used to amplify cDNA following reverse transcription (Table 2). 

### 2.8. Crude Synaptosome Preparation and Immunoblotting

Tissue-specific protein was homogenized at 0 °C in 50 volumes (*w*/*v*) of sucrose buffer 0.32 M sucrose buffer, 4.2 mM HEPES buffer (pH 7.4), 0.1 mM CaCl_2_, 1 mM MgCl_2_ plus protease inhibitor and phosphatase inhibitor]. Samples were centrifuged at 1000× *g* for 10 min at 4 °C to discard large debris. The supernatant was transferred to a fresh centrifuge tube and then centrifuged at 10,000× *g* for 20 min at 4 °C. The supernatant was removed, and the pellet was gently resuspended in 40 volumes (*w*/*v*) of hypotonic buffer [20 mM Tris-HCl (pH 6.0), 0.1 mM CaCl_2_, 1 mM MgCl_2_, 1% Triton X-100 plus protease inhibitor and phosphatase inhibitor]. The suspension was incubated for 20 min at 4 °C with rotation to collect the crude synaptosome.

From here a 1:1 volume (*w*/*v*) of 2× RIPA buffer [300 mM NaCl, 2.0% NP-40, 0.1% SDS (sodium dodecyl sulphate), 100 mM Tris-HCl (pH 8.0), protease inhibitor and phosphatase inhibitor] was added into the crude synaptosome suspension. Centrifugation at 14,800 rpm at 4 °C for 20 min, the supernatant was collected and Laemmli sample buffer was added. Samples were boiled at 100 °C for 5 min for Western blots [31].

Western blots were quantified with GeneTools from SYNGENE according to the manufacturer’s instruction. Histone H3 modification changes were calculated by dividing H3K4me, H3K27me, H3K9ac, H3K14ac, H3K18ac and H3K27ac by histone H3. Histone H3 modification changes were represented using the relative fold change of expression and were determined by all groups versus WT vehicle treatment (defined as 1.0).

### 2.9. Statistical Analysis

Control and experimental groups were randomly assigned by cage. All experiments contained littermates and non-littermates, which were both randomly assigned to control and experimental group. Sample size for experiments were based upon previously published experiments [32]. Statistical analyses were performed using Prism software (GraphPad, La Jolla, CA, USA). Data were shown as means ± SEM. The data were analyzed using two-way ANOVAs with one between-subject factor (genotype: WT or GHRH-KO) and one dependent variable, followed by Tukey post hoc tests. *p* < 0.05 was considered significant, * *p* < 0.05 and ** *p* < 0.01.

### 2.10. Study Flowchart

The Study Flowchart is as follows



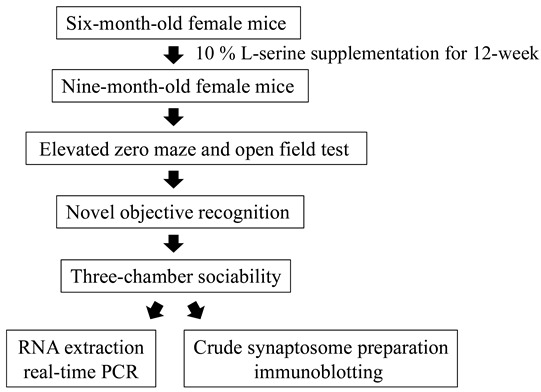



## 3. Results

### 3.1. L-Serine Intervention Decreased Anxiety-Related Behavior

Anxiety symptoms commonly occur in adolescent females, leading to an increased risk in psychosocial functioning [33], and there are few studies on emotional behaviors and cognitive functions in female GHRH-KO mice. Thus, nine-month-old female KO mice were used to investigate whether GH affects anxiety-related behaviors using OFT and EZM. Increased center time of OFT and open arms time of EZM are the parameters that reflect an anxiolytic effect [28]. KO and WT mice spent an almost equal amount of the number of entries and time in the center of entire OFT (Figure 1B,C). In addition, we recorded no significant difference in the number of entries into open EZM arms between KO and WT mice (Figure 1D), however, the amount of time spent in the open arms of the EZM significantly increased in the mice with a loss of GH (Figure 1E). These data indicate that GH deficient mice present fewer anxiety symptoms.

Genotype or drug intervention affects anxiety-like behavior with increased entry time in the open regions of the EZM and in the center of the OFT [34]. A previous study showed that 10% L-serine supplementation increased L-serine levels in the brain [35] and positively affected metabolism [36], thus we sought to investigate whether 12 weeks of 10% L-serine treatment influences anxiety-like disorders in WT and GH deficiency-induced metabolic changed KO mice. As shown in Figure 1A, additional L-serine in the water did not significantly change water intake in WT and KO mice. However, we found that the number of entries into the center of the OFT increased 2-fold and the number of entries into the open arms of the EZM increased 25% in both genotypes following oral administration of L-serine, respectively (Figure 1B,D). WT mice subjected to L-serine intervention spent more than 60% of the trial time in the center of the OFT and more than 33% of the total trial time in the open arms of the EZM. These results were similar to those observed with L-serine-treated KO mice (Figure 1C,E). These data indicate that L-serine intervention has an anxiolytic effect with decreased anxiety-like behavior.

### 3.2. L-Serine Intervention Improved Novel Object Recognition in GHRH-KO Mice

The function of gonadal hormones is related to components of memory and gender is an important consideration to assess the working and spatial memory [37]. The NOR is a useful tool to study short- and long-term memory with various inter-trial intervals [38]. The preference for a novel object reflects the existing memory of a familiar object in rodents. To understand the functions of GH and L-serine intervention in memory, KO and WT mice were exposed to two identical objects in the training session. A four-hour inter-trial interval was selected to evaluate short-term recognition memory. In the novel object recognition trial, the mice were exposed to two identical objects in the training session and given a new object that replaces one of the two familiar objects during the testing session. In contrast with WT, KO mice failed to recognize the novel object with the similar parameters of frequency and exploring time (Figure 2A,B). However, oral administration of L-serine in KO mice resulted in a 2-fold increase in cognitive frequency (average from 23 to 45) and in spending time (17% with vehicle treatment versus 32% with L-serine treatment) in novel object exploration during the entire NOR trial (Figure 2A,B).

### 3.3. L-Serine Intervention Enhanced Sociability in GHRH-KO Mice

Autism in women is commonly misdiagnosed by general expectancy biases, such as shyness or anxiety [39]. Sociability is an important early marker for autism and related neurodevelopmental disorders [40]. To understand whether a loss of GH signaling is linked to autism-like diseases, KO and WT mice were tested relative to exploration of a side chamber containing an unfamiliar conspecific in a cage, or an empty side chamber. We determined that WT mice showed a marked preference for exploring the chamber containing the stranger mice compared with the empty chamber (Figure 3A). Further, a clear preference for spending time with the stranger mice was observed in the WT group (Figure 3B). However, the significant preference of sociability with the stranger mice was attenuated in KO mice (Figure 3A,B). KO mice exhibited an autism-like behavior compared to WT mice. Oral administration of L-serine resulted in KO mice exploring and spending more time with the stranger mice (Figure 3A,B), indicating a beneficial effect of L-serine treatment.

### 3.4. DNA Methylation Was Slightly Altered by Oral Administration of L-Serine

Dysfunctions of Dnmt and Tet, which are two important enzymes in the regulation of DNA methylation patterns, have been noted in behaviors related to psychiatric disorders [2,3]. A study showed that the mRNA level of DNMTs is associated with enhanced activities of DNMTs, which contributes to regulation of global DNA methylation in GH-deficient Ames dwarf mice [41]. We thus tested the expression of Dnmt and Tet genes in isolated cerebral cortex and hippocampus tissues using reverse transcriptase–polymerase chain reaction analysis. The mRNA expressions of *Dnmts* and *Tet1*/*3* in the brain and *Tet2* in the cerebral cortex were similar in both genotypes (Figure 4). However, the mRNA level of *Tet2* was reduced 2-fold in the hippocampus of KO mice compared to WT controls (Figure 4D). In addition, oral administration of L-serine slightly decreased *Tet1* and *Tet2* in the hippocampus and cerebral cortex of WT, and *Tet2* in the cerebral cortex and *Tet1* in the hippocampus of KO mice (Figure 4B,D). We found that oral administration of L-serine slightly increased expression of *Dnmt1* in the hippocampus, but not in the cerebral cortex in both genotypes (Figure 4A,C). These data suggest that L-serine may influence global DNA methylation and demethylation in a tissue-specific manner.

### 3.5. Oral Administration of L-Serine Changed Profiles of Histone H3 Modification

Histone H3 methylation and acetylation on multiple residues play important roles in modification of synaptic function and formation of long-term memory in mammals [42]. Using histone H3, we profiled the methylation patterns of lysine 4 and 27 (H3K4me and H3K27me) and acetylation of lysine 9, 14, 18 and 27 (H3K9ac, H3K14ac, H3K18ac and H3K27ac) in the crude *synaptosome* of the cerebral cortex and hippocampus from WT and KO mice. As shown in Figure 5, under vehicle treatment, H3 methylation and acetylation were not significantly different between KO mice and WT controls. Oral administration of L-serine markedly increased H3K4me, H3K9ac, H3K14ac and H3K18ac in the hippocampus (Figure 5A,B), and induced acetylation of H3K14 in the cerebral cortex in both genotypes (Figure 5C,D). These data suggest that L-serine intervention modulated the histone H3 modification patterns in different brain regions.

### 3.6. Epigenetic Modification-Regulated Genes Were Changes by L-Serine to Improve Behavioral Deficits

Our data indicated that L-serine intervention attenuated anxiety-like and autism-like behaviors, and improved impairment of short-time recognition memory in KO mice. Here, we sought to investigate whether L-serine intervention-mediated epigenetic modifications could be associated with behavioral changes. We examined mRNA levels of anxiety-, autism- and memory-related genes, expression of which are involved in DNA methylation and histone modifications. Expression of *BDNF*, *grm3*, *foxp1*, *shanks3* and *auts2* was markedly induced in the hippocampus of KO mice with L-serine treatment (Figure 6). Furthermore, L-serine administration markedly reduced mRNA level of *marcksl1* in the cerebral cortex of WT and in the hippocampus in both genotypes (Figure 6F). These data suggest that oral administration of L-serine changed epigenetic modification-related gene expression, which could contribute to improvement of behavioral deficits.

## 4. Discussion

### 4.1. GH Deficiency Induced Cognitive Deficit and Autism-Like Behavior

Aging-related cognition and memory gradually decline in rodents and elderly humans [43]. GH deficiency/resistance has beneficial effects in slowing aging of physical, behavioral, and cognitive attributes. Ames dwarf [44] and GHR-KO mice [45] show a delay in cognitive aging via increased retention in the inhibitory avoidance task. Age-related cognitive dysfunction is observed in heterozygous GHRH-KO mice at 5 months old, while in homozygous GHRH-KO mice the dysfunction is delayed until 12 months of age [14], suggesting that disrupted GHRH/GH axis delays aging-related impairment of cognition.

NOR is widely used to assess non-spatial cognitive deficits dependent on spontaneous innate preference of rodents to investigate novel objects [38]. Normal rodents spend more time exploring novel objects compared with familiar objects, which is consistent observations testing WT mice. Our data show that 9-month-old GHRH-KO mice have significantly decreased time and frequency of exploring novel objects, however, age-matched WT showed a preference for exploring novel objects. These data indicate that GH deficiency attenuated short-term non-spatial memory. Patients with childhood-onset growth hormone deficiency show cognitive dysfunction, particularly in memory performance, however, this memory impairment can be counteracted by growth hormone substitution [46]. Intriguingly, Ames dwarf mice have significantly enhanced short-term spatial memory compared to WT [47]. GHR-KO mice show increased total neuron cell density in the cortex [12] at 24 months of age, as well as improved spatial learning and memory in the Morris water maze compared with WT controls [13]. These data imply that GH may play different roles in spatial and non-spatial memory.

Sociability is defined as the tendency to approach and remain proximal to an unfamiliar con-specific (stranger), compared with avoidance of the stranger mouse by remaining in the center chamber or exploring an equally novel chamber devoid of another mouse [40]. Normal mice sociability involves spending more time with a stranger mouse than in the empty chamber [40]. However, we found that GHRH-KO mice lost preference for a stranger mouse, showing autism-related behavior.

### 4.2. L-Serine Intervention Ameliorates Cognitive Deficit and Autism-Like Behaviors

L-serine serves as a precursor for the neuromodulators glycine and D-serine, which are tightly associated with neurological functions in mammals [22]. However, excess D-serine induces nephrotoxicity in rodents and humans [48]. We thus used the L-enantiomer of serine to investigate possible efficacy in cognitive deficit and autism-like phenotypes.

L-serine provides nucleotide precursors for cell proliferation in the central nervous system (CNS) and significantly affects dendritogenesis and axon length in neurons [20]. L-serine acts as a neuroprotector on the ischemic-reperfused brain and on hippocampal neurons with hypoxia- or glutamate treatments, which may be mediated by activating glycine receptors [49]. The serine racemase enzyme may also serve to change the chirality of other molecules; mice dosed with L-BMAA showed a small pool of the D-enantiomer of BMAA in their liver, and nonhuman primates orally dosed with L-BMAA showed increases of the D-BMAA in their plasma and CNS [50]. In terms of serine, patients with GRIN2B-related pediatric encephalopathy showed remarkably improved communication, social and cognitive functions when dosed with L-serine for an extended period suggesting that L-serine may enhance glutamatergic neurotransmission and correction of excitatory or inhibitory neurotransmitter imbalance linked to a wide range of neurological disorders [22]. Our data show that GHRH-KO mice with L-serine administration have an increased preference for novel objects, and increase the duration and frequency of exploring time in the presence of a stranger mouse rather than remaining in the center or empty chamber.

### 4.3. L-Serine Improves GH Deficiency-Induced Behavioral Deficits Via Epigenetic Modifications

Epigenetic changes, including DNA methylation and histone modifications, have been implicated in behavioral adaptations that rely on learning processes as well as in the subsequent storage of putative memory traces in the central nervous system [42]. *Dnmt1/3α* double knockout mice have impaired spatial learning and memory in Morris water maze and contextual fear conditioning [2]. We found that L-serine administration slightly increased the expression of Dnmt1 and significantly activated the expression of H3K4me, H3K9ac and H3K18ac in the hippocampus of both genotypes. These data are consistent with the widely accepted concept that the hippocampus is an essential part of the memory system involving storing memories, spatial processing and navigation compared with perirhinal cortex or the medial prefrontal cortex [51]. Additionally, these data suggest that L-serine intervention may ameliorate neurological disorders which result from epigenetic changes.

Increased histone H3 acetylation is associated with activated gene transcription. We, therefore, examined cognitive and autism-related gene expressions, in particular the genes that are sensitive to histone H3 modifications. *BDNF* is activated by DNA methylation and plays important roles in synaptic plasticity and the maintenance of memory [5]. Additionally, histone acetylation-mediated *BDNF* improves long-term memory in male mice [52]. Group II metabotropic glutamate receptors 2 are encoded by Grm2 and Group II metabotropic glutamate receptors 3 which are encoded by Grm3. These are regulated by DNA methylation in promoter regions and are involved in cognitive and emotional processes which have been linked with a number of neuropsychiatric conditions [53]. *grm2* and *grm3* single knockout mice reduce spatial working memory which is further decreased in *grm2/3* double knockout mice [54]. Foxp1 is regulated by epigenetic mechanisms and functions in memory development and maintenance [6]. Brain-specific deleted *foxp1* impaired short-term memory and causes autistic-like behavior in mice [55]. Shank3 is regulated by DNA methylation [7]. Deletion/mutation of the *Shank3* in humans and mice is related to autism spectrum disorder [56]. Expression of Auts2 is controlled in neurons by DNA-methylated enhancer regions [8]. Imbalanced expression of Auts2 affects neurodevelopment and neurological function, which results in autism-like and other neurological disorders in zebrafish [57]. Our data indicate that expressions of *BDNF*, *Grm3*, *Foxp1*, *Auts2* and *Shanks3* in the hippocampus are significantly induced by L-serine administration in GHRH-KO mice via changed expression of epigenetic markers, which may ameliorate impairment of memory and autism-related behavior.

### 4.4. L-Serine Intervention Has an Anxiolytic Effect on Behavior

Anxiety symptoms commonly occur in adolescent females, which are associated with increased risk in psychosocial functioning in adults [33]. Clinic studies report that anxiety disorders at early ages are more common in girls than boys, suggesting sex dependence in anxiety [58]. The increased amount of time spent in the open arms of the EZM implied a lower anxiety phenotype in GHRH-KO females, which is consistence with the GHRH-KO males. These data suggest that impaired GHRH/GH axis signaling may have beneficial effects on anxiety-like phenotype. In addition to genetic factors, epigenetic changes including DNA methylation and histone modifications are associated with anxiety disorders. Deleted *Tet1* in NAc increased the anti-anxiety-like effect in mice [3]. Our data indicate that L-serine administration slightly decreases the expression of Tet in the cerebral cortex and hippocampus of both genders, which may contribute to a decrease in anxiety-like phenotype. Overexpressed Marcksl1 induces anxiety-like behaviors in Marcksl1 Tg C57BL/6N mice, however, decreased Marcksl1 ameliorates anxiety-like behaviors in mice [59]. Downregulated Marcksl1 by histone deacetylase inhibitor [9] suggests that Marcksl responds to epigenetic modification. Our data show a significant decrease in hippocampal *Marcksl1* in GHRH-KO and WT mice with L-serine dosing, suggesting that L-serine has an anxiolytic effect on behavior via regulation of histone modification-mediated Marcksl1 expression in mice.

## 5. Summary

In our study, disruption of GH signaling decreased anxiety symptoms, however, caused impairment of short-term object recognition memory and autism-like behaviors in adolescent females. L-serine administration exerted anxiolytic effects in mice and ameliorated behavioral deficits in KO mice. Furthermore, L-serine treatment upregulated the gene expression of H3K4me, H3K9ac, H3K14ac and H3K18ac in the hippocampus and H3K4me in the cerebral cortex in both KO and WT mice. The changed mRNA level of *BDNF*, *grm3*, *foxp1*, *shank3*, *auts2* and *marcksl1* are involved in anxiety-, cognitive- and autism-like behaviors.

## Figures and Tables

**Figure 1 biomedicines-11-00104-f001:**
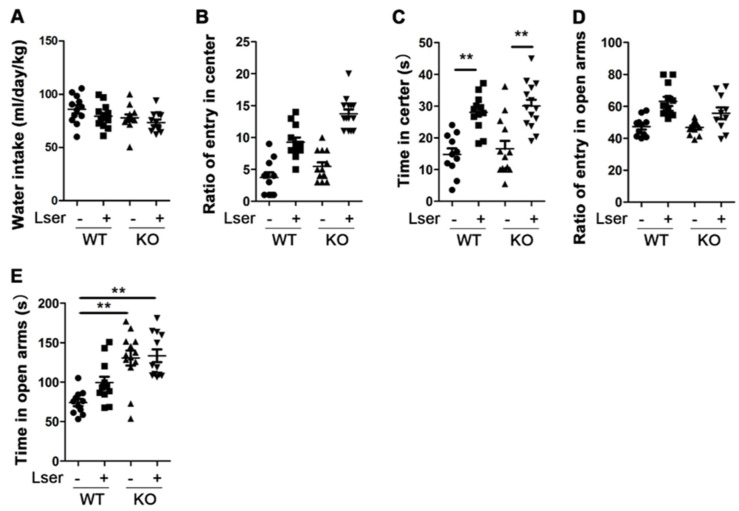
Oral administration of L-serine reduced anxiety-related behavior in mice. GHRH-KO mice and age-matched wild type controls were subject to oral administration of L-serine for 12 weeks before EZM and OFT. (**A**) Water intake. (**B**) Ratio of entry in the center during the entire OFT time. (**C**) Time spent in the center during the entire OFT time. (**D**) Ratio of entry in open arms during the entire EZM trial. (**E**) Time spent in open arms during the entire EZM trial. *n* = 11–14 mice per group. Data are means ± SEM. ** *p* < 0.01.

**Figure 2 biomedicines-11-00104-f002:**
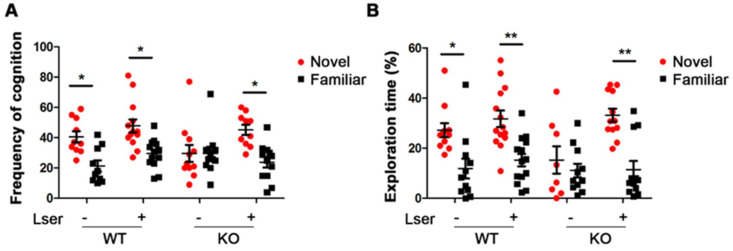
Oral administration of L-serine improved novel object recognition (NOR) in GHRH-KO mice. NOR was performed with GHRH-KO mice and age-matched wild type controls upon L-serine intervention for 12 weeks. (**A**) Frequency of exploration in novel and familiar objects during the entire NOR trial time. (**B**) Percentage of time spent investigating novel and familiar objects during the entire NOR trial time. *n* = 11–14 mice per group. Data represent means ± SEM. * *p* < 0.05, ** *p* < 0.01.

**Figure 3 biomedicines-11-00104-f003:**
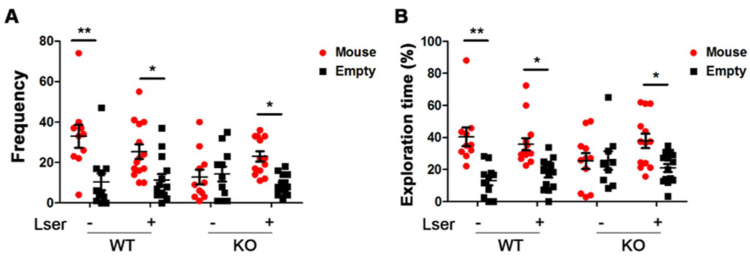
Oral administration of L-serine attenuated autistic-like behavior in GHRH-KO mice. Three-chamber sociability was conducted with GHRH-KO mice and age-matched wild type controls upon L-serine intervention for 12 weeks. (**A**) Frequency of exploring in the empty cage and strange mouse cage during the entire sociability trail time. (**B**) Percentage of exploration time spent in the empty cage and strange mouse cage during the entire sociability trail time. *n* = 11–14 mice per group. Data represent means ± SEM. * *p* < 0.05, ** *p* < 0.01.

**Figure 4 biomedicines-11-00104-f004:**
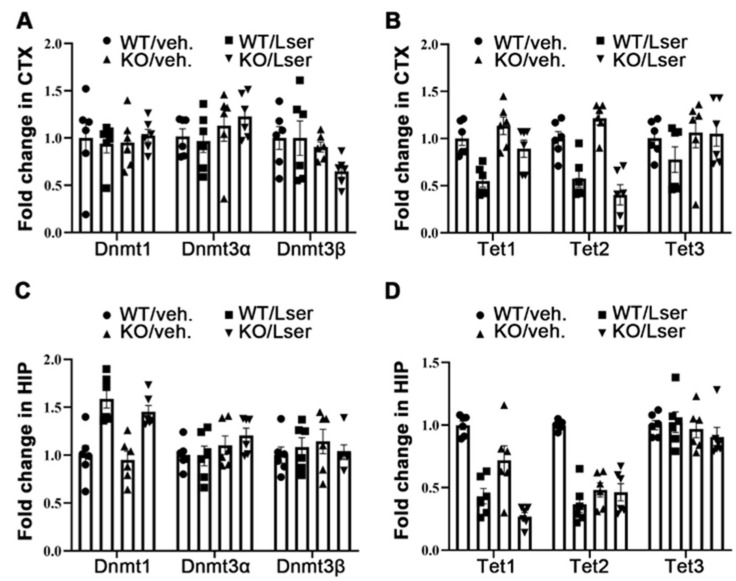
Oral administration of L-serine regulated mRNA level of Dnmt and Tet in GHRH-KO mice and wild type controls. GHRH-KO mice and age-matched wild type controls were subject to L-serine oral administration for 12 weeks. The mice were euthanized and tissues were collected for RNA extraction. (**A**) Expression of *Dnmt* in the cerebral cortex. (**B**) Expression of *Tet* in the cerebral cortex. (**C**) Expression of *Dnmt* in the hippocampus. (**D**) Expression of *Tet* in the hippocampus. *n* = 6 mice per group. Data represent means ± SEM.

**Figure 5 biomedicines-11-00104-f005:**
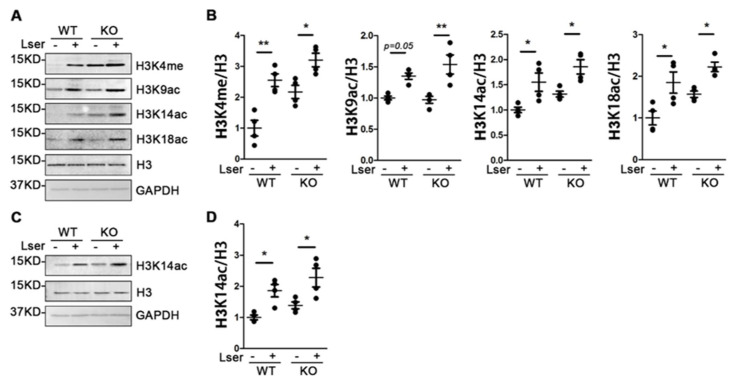
Enriched profile of histone H3 modifications in GHRH-KO mice and wild type controls with L-serine intervention. GHRH-KO mice and age-matched wild type controls were subject to L-serine oral administration for 12 weeks. The mice were euthanized and tissue-specific crude synaptosome was extracted for immunoblot analyses. (**A**) Immunoblots of crude synaptosome in the hippocampus are shown for H3K4me, H3K9ac, H3K14ac and H3K18ac. (**B**) Quantification of hippocampal H3K4me, H3K9ac, H3K14ac and H3K18ac activation with L-serine supplementation for 12 weeks in both genotypes. (**C**) Immunoblot of crude synaptosome in the cerebral cortex is shown for acetylation of H3K14. (**D**) Quantification of cerebral cortical H3K14ac activation with L-serine supplementation for 12 weeks in both genotypes. *n* = 4 mice per group. Data represent means ± SEM. * *p* < 0.05, ** *p* < 0.01.

**Figure 6 biomedicines-11-00104-f006:**
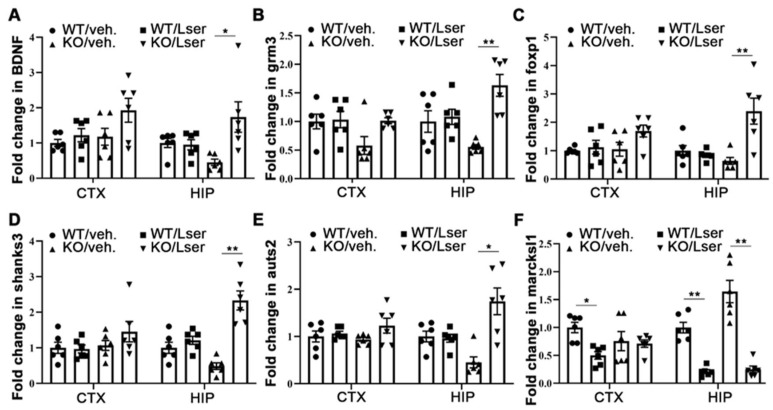
Oral administration of L-serine regulated gene expression in hippocampus and cerebral cortex of GHRH-KO mice and wild type controls. GHRH-KO mice and age-matched wild type controls were subject to L-serine oral administration for 12 weeks. The mice were euthanized and tissues (CTX, cerebral cortex and HIP, hippocampus) were collected for RNA extraction. (**A**) Expression of *BDNF*. (**B**) Expression of *grm3*. (**C**) Expression of *foxp1*. (**D**) Expression of *shanks3*. (**E**) Expression of *auts2*. (**F**) Expression of *marcksl1*. *n* = 6 mice per group. Data represent means ± SEM. * *p* < 0.05, ** *p* < 0.01.

**Table 1 biomedicines-11-00104-t001:** Antibody and chemicals.

No.	Protein Name	Monoclonal or Polyclonal	Host Species	Commercial Supplier	Catalogue Number
1	H3K4me	polyclonal	Rabbit	Abcam	ab272143
2	H3K27me	monoclonal	Rabbit	Abcam	ab192985
3	H3K9ac	Monoclonal	Rabbit	Cell Signaling Technology	#9649
4	H3K14ac	Monoclonal	Rabbit	Cell Signaling Technology	#7627
5	H3K18ac	Monoclonal	Rabbit	Cell Signaling Technology	#13998
6	H3K27ac	Monoclonal	Rabbit	Cell Signaling Technology	#8173
7	H3	Monoclonal	Rabbit	Cell Signaling Technology	#4499
8	GAPDH	Monoclonal	Mouse	Cell Signaling Technology	#97166

**Table 2 biomedicines-11-00104-t002:** Primers used to amplify cDNA following reverse transcription.

Name	Forward Primer (5′-3′)	Reverse Primer (5′-3′)
Dnmt1	TTGAAACTTCACCTAGTTCCGTGGC	CTGCAGCACCACTCTCTGTGTCTAC
Dnmt3α	GAAAGGGTATGGGAGTTACATAGAG	CGGTGTGAAATATCGCAGTTTAAG
Dnmt3β	CACACTCTGGAGAAAGCCAGGGTTC	AGTCATTGGTTGTGCGTCTTCGACT
Tet1	GACCCTCATAAGCAGAGAGGAAAA	TCTTCATTTCCAAGTCGACAGTCT
Tet2	GGAGCAGAAGGAAGCAAGATGG	ATGAATCCAGCAGCACCGTCC
Tet3	AAATGCTCGTGAAGGAACGGG	CTGCCTTGAATCTCCATGGTACAC
*BDNF*	CGAGATCGGGGCTGGAGA	GGTCATCACTCTTCTCACCTGG
Grm3	TCATTGGCGGTTCGTACAGC	ACTGGTGGAGGCGTAGCTTAT
Foxp1	CACCTCAGGTTATCACTCCTCA	AGCTGCAACTGTTCCTGTTGT
Shank3	CAACATGGGTGCTCAGAATGC	ACGACATCGGAGTCTTTGTGG
Auts2	TCAGCCACTCACACCACTACA	AGCCCTTGGATTCTCCGCT
Marcksl1	GCCAACGGACAGGAGAATGG	CTCGATGGCATCACCAGTAGC
β-actin	TCTTTGCAGCTCCTTCGTTGCC	CTGACCCATTCCCACCATCACAC
GAPDH	CCTGGAGAAACCTGCCAAGTATGATG	AAGAGTGGGAGTTGCTGTTGAAGTC

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
