# Peer review of "L-Serine Influences Epigenetic Modifications to Improve Cognition and Behaviors in Growth Hormone-Releasing Hormone Knockout Mice"

_biomedicines, 2022, doi:10.3390/biomedicines11010104_

Round 1

Reviewer 1 Report

Thank you for the opportunity to read this well written, clear and concise study. It was interesting and a pleasure to read.

The introduction is very complete and allows the reader to know the main topic of the research, informs about the purpose and importance of the work in the clinical field, and also answers the question posed in the scientific context. It includes previous works on the subject in question and makes clear the aspects to be detailed, which constitutes the object of the proposed investigation.

The "methods" section is one of the most fundamental sections of a scientific article with these characteristics. However, some aspects should be reviewed to give the manuscript a higher quality:

- It is necessary a first subsection in which the type of study is specified, the approval of the Ethics Committee of the corresponding institution.

- Add a sample size calculation, referring to the literature and in sufficient detail to allow replication.

- Include the Flowchart showing the assignment of study participants.

In the Results section, it is necessary to start with the sociodemographic data of the sample to provide background information to the reader and explain in detail the main characteristics of the sample.

A final conclusion section should be included that summarizes the main results of the investigation.

Author Response

Dear editors and reviewers,

We are pleased to submit the revised draft of our manuscript, “L-serine influences epigenetic modifications to improve cognition and behaviors in growth hormone-releasing hormone knockout mice” (biomedicines-2108521), to Biomedicines. We appreciate the time and effort dedicated by the editorial staff and reviewers. The comments provided were valuable and helped us refine our paper. As such, we have made several revisions to the manuscript based on the suggestions given. Changes to the manuscript are highlighted.

Below are our point-by-point responses to the reviewers’ comments.

Response to Reviewer 1

Thank you for your insightful comments and suggestions. Please find the answers to each of your questions below.

  1. It is necessary a first subsection in which the type of study is specified, the approval of the Ethics Committee of the corresponding institution.

Response: Yes, it is a necessary information to the entire study. So please find the added information in lines 76-80.

  1. Add a sample size calculation, referring to the literature and in sufficient detail to allow replication.

Response: Thank you for your close reading of our paper. This is an important point, so we have added a literature in line 165.

  1. Include the Flowchart showing the assignment of study participants.

Response: Thank you for this suggestion. We added a study flowchart in line 171.

  1. In the Results section, it is necessary to start with the sociodemographic data of the sample to provide background information to the reader and explain in detail the main characteristics of the sample.

Response: A: Thanks for your suggestions. We added the sociodemographic data in the result section to explain the background of samples in lines 209-210 and lines 230-231.

  1. A final conclusion section should be included that summarizes the main results of the investigation.

Response: Thank you for this suggestion. A summary section was added in lines 412-419.

Response to Reviewer 2

Aging-related cognition and memory gradually decline in rodents and elderly humans. GH deficiency/resistance has beneficial effects in slowing aging of physical, behavioral, and cognitive attributes. Ames dwarf and GHR-KO mice show a delay in cognitive aging via increased retention in the inhibitory avoidance task. Age-related cognitive dysfunction is observed in heterozygous GHRH-KO mice at 5 months old, while in homozygous GHRH-KO mice the dysfunction is delayed until 12-months of age, suggesting that disrupted GHRH/GH axis delays aging-related impairment of cognition. Overall, the show a significant decrease in hippocampal Marcksl1 in GHRH-KO and WT mice with L-serine dosing, suggesting that L-serine has an anxiolytic effect on behavior via regulation of histone modification-mediated Marcksl1 expression in mice.

Although animals’ models of psychiatric diseases are of limited value for clinicians, the study is very well conducted and deserves publication.

Response: Thank you for approval our study.

Additionally, we have thoroughly proofread the manuscript to eliminate any remaining grammatical and spelling errors.

We look forward to your response regarding our submission. Please do not hesitate to contact us if there are any further questions or comments.

Sincerely,

Fang Zhang and Liou Y. Sun

Reviewer 2 Report

Aging-related cognition and memory gradually decline in rodents and elderly humans. GH deficiency/resistance has beneficial effects in slowing aging of physical, behavioral, and cognitive attributes. Ames dwarf and GHR-KO mice show a delay in cognitive aging via increased retention in the inhibitory avoidance task. Age-related cognitive dysfunction is observed in heterozygous GHRH-KO mice at 5 months old, while in homozygous GHRH-KO mice the dysfunction is delayed until 12-months of age, suggesting that disrupted GHRH/GH axis delays aging-related impairment of cognition. Overall, the show a significant decrease in hippocampal Marcksl1 in GHRH-KO and WT mice with L-serine dosing, suggesting that L-serine has an anxiolytic effect on behavior via regulation of histone modification-mediated Marcksl1 expression in mice. 

Altough animals models of psychiatric diseases are of limited value for clinicians, the study is very well conducted and deserves publication.

Author Response

(The authors gave the same response as above.)

Round 2

Reviewer 1 Report

The authors have made all the suggested modifications, therefore the manuscript is accepted.